# Predicting Corrosion Delamination Failure in Active Implantable Medical Devices: Analytical Model and Validation Strategy

**DOI:** 10.3390/bioengineering9010010

**Published:** 2021-12-31

**Authors:** Adrian Onken, Helmut Schütte, Anika Wulff, Heidi Lenz-Strauch, Michaela Kreienmeyer, Sabine Hild, Thomas Stieglitz, Stefan Gassmann, Thomas Lenarz, Theodor Doll

**Affiliations:** 1Department of Engineering, Jade University of Applied Sciences, 26382 Wilhelmshaven, Germany; adrianonken@gmail.com (A.O.); helmut.schuette@jade-hs.de (H.S.); lenz-strauch@jade-hs.de (H.L.-S.); stefan.gassmann@jade-hs.de (S.G.); 2Department of Otolaryngology, Hannover Medical School MHH, 30625 Hannover, Germany; Wulff.Anika@mh-hannover.de (A.W.); Kreienmeyer.Michaela@mh-hannover.de (M.K.); Lenarz.Thomas@mh-hannover.de (T.L.); 3Institute of Polymer Chemistry, Johannes Kepler University, 4010 Linz, Austria; sabine.hild@jku.at; 4Department of Microsystems Engineering—IMTEK, University of Freiburg, 79110 Freiburg, Germany; thomas.stieglitz@imtek.uni-freiburg.de; 5Fraunhofer Institute of Toxicology and Experimental Medicine ITEM, 30625 Hannover, Germany

**Keywords:** AIMD, corrosion-triggered delamination, PDMS, moving boundary diffusion, body fluids

## Abstract

The ingress of body fluids or their constituents is one of the main causes of failure of active implantable medical devices (AIMDs). Progressive delamination takes its origin at the junctions where exposed electrodes and conductive pathways enter the implant interior. The description of this interface is considered challenging because electrochemically-diffusively coupled processes are involved. Furthermore, standard tests and specimens, with clearly defined 3-phase boundaries (body fluid-metal-polymer), are lacking. We focus on polymers as substrate and encapsulation and present a simple method to fabricate reliable test specimens with defined boundaries. By using silicone rubber as standard material in active implant encapsulation in combination with a metal surface, a corrosion-triggered delamination process was observed that can be universalised towards typical AIMD electrode materials. Copper was used instead of medical grade platinum since surface energies are comparable but corrosion occurs faster. The finding is that two processes are superimposed there: First, diffusion-limited chemical reactions at interfaces that undermine the layer adhesion. The second process is the influx of ions and body fluid components that leave the aqueous phase and migrate through the rubber to internal interfaces. The latter observation is new for active implants. Our mathematical description with a Stefan-model coupled to volume diffusion reproduces the experimental data in good agreement and lends itself to further generalisation.

## 1. Introduction

Neural prostheses emit electrical signals directly into nerve paths restoring lost neural functions. In this way, neural faults and dysfunctions can be treated. According to the EU Medical Device Regulation, neural prostheses are active implantable medical devices (AIMDs) which belong to the highest risk Class III of medical devices. This study addresses neural prostheses, which are implanted permanently into the human body and exposed to its fluids [1,2,3]. The cochlear implant serves as a blueprint and frame of reference for the developed specimen in this study, as it is the most commonly implanted neural prosthesis [4]. In this kind of prosthesis, an electrode array forms the interface between the implant and the nervous system. The electrodes and the connecting wires—so-called leads—are commonly encapsulated in silicone rubber, which serves as elastic, isolating and bio inert carrier material [3].

Our study focuses on the corrosion-triggered delamination of the silicone rubber from the encapsulated metal of the electrode with its lead or an encapsulated implantable pulse generator (IPG), leading to leakage and electrical dysfunctions due to the change of impedance and isolation properties. The driving force of this delamination process is assumed to be the diffusion of molecules of the body fluids into the interface leading to weakening of the adhesion between electrode and encapsulation and eventually leading to corrosion.

Polydimethylsiloxane (PDMS) is one of the standard silicon rubbers, which are often used in a wide range of medical applications. PDMS is known to be nontoxic, easy to process and shows quite favourable mechanical and electrical properties. Together with its hydrophobic nature and long-term stability, PDMS is considered an ideal encapsulation of sensitive electronic circuits of AIMDs such as the cochlear implant (CI) [5]. In spite of this hydrophobic nature and its use for electrical insulation, the water absorption and diffusion are still a concern in several studies, as they become a factor in the long term [6,7,8]. This could result in corrosion and failure of embedded leads and electronic circuits. A better understanding of these diffusion processes may lead to a better prediction of aging and failure of such devices.

On that account, a collection of studies dealing with the modelling of sorption and transport processes of water vapour exists. High individuality of different PDMS blends and its filler components make each study quite specific depending on manufacturer and product and becoming more unique when the PDMS is used in a metal-polymer compound like implantable neural electrodes, with varying geometry, composition depending on manufacturer and fabrication method [9,10,11,12].

Furthermore, all existing studies strictly focus on diffusion processes in bulk, not giving credit to transport and aging processes in the interface of such polymer-metal joints (Figure 1). This might be due to the interface being only a few nanometres thick. With progressing miniaturization of medical devices, respectively electrical devices in general, the thickness of encapsulating materials decreases further and further. Consequently, the relation between interface and the material volume begins to turn and the impact of the interface gains more importance [13]. Nevertheless, there are very few studies investigating these interface processes. Not only is there a lack of mathematical understanding of the transport and aging mechanisms at the interface of metal-polymer joints, but in interface diffusion processes in general. Most developed mathematical methods strongly focus on diffusion in bulk, not crediting the importance of the events taking place at the interface. 

This may be ironic due to the fact that a major failure mechanism is the delamination of the PDMS mantle of the platinum wire as observed in cochlear implants. This delamination mainly arises at the part of the electrode where it has its contact [14]. At these contacts, the body fluids begin to corrode the platinum alloy with the corrosion migrating further into the electrode. It can be assumed that this migration takes place in the interface of the polymer-metal joint, with the adhesion and the geometry of the edge, where the three phases (i.e., body fluid, metal and polymer) meet. In order to investigate the processes at the interface, a mathematical model describing the different correlating processes of diffusion, corrosion and delamination was developed in this study and validated by a simple experimental method. For the purpose of acceleration of the degradation process, the platinum was exchanged for copper as there are many ways to corrode copper chemically in short amounts of time, whilst the interface energies of both metals with PDMS remain similar. Specimens that are easy to fabricate were developed allowing a visual observation of the interface diffusion. Crediting the impact of diffusion in bulk, which is still assumed as a not deniable factor, additional experiments for the bulk diffusion processes in a special experimental setup were performed. 

A mathematical interpretation based on the works of Crank and Goodman was developed and is introduced in Section 3.1. This model was eventually validated by our experimental data.

## 2. Materials and Methods

### 2.1. Rationale of the Study

Manufacturers of cochlear implants award a product lifetime up to 30 years. Implant failure is an inherent ubiquitous risk [5]. A study of the German legal authority of medical devices (BfArM) researched failure mechanisms of cochlear implants over 23 years. The reason that led to reimplantation was device failure. Rate of re-implantation of silicone rubber-based electrode arrays was 7.3% (Nucleus, Cochlear, Inc. Sydney, NSW, Australia) and 5.3% (Clarion, Advanced Bionics, Valencia, CA, USA), respectively. The study names the four most frequent failure mechanisms: impact, leakage, electrical dysfunction and abnormal electrode arrangement. Out of those four failure mechanisms, leakage was identified as the most common type of error [15].

Our study focuses on corrosion-triggered delamination of the PDMS from the metal electrode, leading to leakage and electrical dysfunctions due to the change of impedance and isolation properties. The driving force of this delamination process is assumed to be the diffusion of molecules of the body fluids into the interface, leading to corrosion weakening the adhesion between electrode and encapsulation. 

The main cause of failure of thin-film components such as microelectrode-arrays is the delamination of layers due to a humid environment. Providing a stable compound between precious metals and polymers can be quite challenging as it is not possible to blend the atoms of each material [16,17]. The most important fracture patterns of adhesive connections were classified in ISO 10365. In this classified case of delamination, humidity penetrates the interface along the cutting edge [18]. Delamination is defined as the separation of two contiguous layers along the interface due to mechanical stress and chemical reactions [19,20]. To counteract the chemical part of delamination, adhesion is often enhanced by machining processes such as surface treatments with oxygen plasma. Alongside plasma treatment there are other approaches to improve adhesion such as chrome metallization [16].

Even with adhesive optimization of those compounds, delamination is still present. This makes further investigations of possible factors influencing delamination essential. A recent study [21], already addressed the impact of thermal stresses in polymers due to shrinkage during polymerization as well as the influence of thermal and mechanical stresses on delamination [21]. This study investigates how diffusion of molecules contributes to corrosion and the weakening of adhesion at the interface. Bulk diffusion is considered a coexisting factor for parallel on-going processes. The combination of these factors impacts the adhesion at the interface and therefore the resulting delamination [22].

One major trend in medical engineering is miniaturization of AIMDs to become minimally invasive and thereby reduce both surgical trauma and scarring in the healing phase after implantation. This miniaturization leads to implant sizes of a few millimetres containing complex integrated circuits in which leakage phenomena are the predominant cause of failure of the mechanism.

Lifetime estimations and valuation of protection against biologic liquids are a big concern in medical approval processes. Due to the small size of cochlear implants, helium leak testing is inapplicable making soak test necessary to examine the resilience of AIMDs against ions and fluids. These tests confirm delamination to be a major failure mechanism [23]. Based on these criteria, the following materials and investigation methods were chosen.

### 2.2. PDMS

PDMS as encapsulation material has a long tradition in AIMDs [3,15]. The PDMS Sylgard-184 (Dow Corning Corporation, Midland, Michigan, United States) has been chosen for this study. It is non-toxic ensuring a high biocompatibility and shows a high hydrophobic nature. It consists of a two-part resin/curing agent of vinyl end-capped oligomeric dimethyl siloxane chains cross-linked with tri-methylated silica and a platinum catalyst. Furthermore, there are additional filler components in the polymer blend, which are unknown by its manufacturer due to its commercial use.

### 2.3. Metal

A copper sputter silica wafer was used as a metal layer. Copper was chosen as it can be corroded within seconds. The manner of this type of corrosion in the experiments showed a sharp visual contrast between corroded and non-corroded metal, allowing easy optical analysis via a camera and a MatLab (MatLAb R2021a, The MATH WORKS, Natik, MA, USA) script. The sputter deposition of the copper layer ensures homogeneous surface properties. Due to the minimization of the observation time, corrosion at the interface due to diffusion becomes the only time-relevant process. Details of copper deposition are described in Section 2.5.

### 2.4. Corroding Agent

Recent studies have only shown ion selective corrosion for the vapour state. Up to this point, diffusion out of the aqueous phase was not considered in most studies. Only a single paper has described diffusion of aqueous bromine in PDMS [24], so far. 

A solution of potassium polysulfide (sulphur liver) (purchased from Sigma-Aldrich Merck, Hamburg, Germany) dissolved in deionized water is chosen as a corroding agent. To only corrode the copper but sparing the Sylgard-184, a 2% concentration of potassium polysulfide was used. Single droplets were placed with a syringe on the edge of the copper and silicone (Figure 2).

Pictures were taken every 5 min to display the time dependent corrosion process (Figure 3). Initial experiments showed fast ion-selective diffusion in PDMS. Immediately after immersing the specimen into the corroding agent, the exposed copper begins to corrode. The corroded copper shows a colour change towards brown/black, while initially the copper under the PDMS remains protected and maintains the original red/orange colour. This sharp contrast allows an automated analysis of the corroded to non-corroded ratio via MatLab script and a calculation of the covered distance by the diffusion front over time. The histogram function of MatLab was used to convert the colours into a black and white picture with as few grey scale levels as possible. First, all occurring colours were assessed in a histogram analysis. A maximum at red colours was chosen for the copper and a maximum at brown/black for the corroded area. A contrast margin of 90 was chosen at which all colours on the left side were set to black and all colours on the right side were set to white. Mathematically, a multiplication with 1 was performed for all colours at a level of up to and equal to 90 and a multiplication with 0 for all values above 90. Black pixels will be counted and subtracted from the overall sum of all pixels to determine the circular area and derive the radius. Analysing the radius over time gives the diffusion length s in pixels over time.

### 2.5. Specimen for Interface Diffusion

To investigate the geometry of the 3-phase-boundary on the interface diffusion, different approaches were followed to form the edge of the PDMS droplet as it marks the point where interface diffusion starts. All attempts of manufacturing a pre-set edge using a mould failed due to the fact that the low viscosity of Sylgard-184 leads to the undermining effects of the polymer. Moreover, there was no way to form an edge without creating mechanic and thermal stresses in the interface, weakening the adhesion of the polymer-metal joint. Investigations of diffusion require a nearly perfect interface without discontinuities or material errors. Therefore, the approach was changed to a more sparing manufacturing process. 

PDMS droplets were set directly on a copper layered wafer, leaving the drop to form naturally. As a continuous surface energy can be assumed, which leads to a contact angle according to the Young equation and the different viscosities in the fluid and solid state, a uniform edge along the outer droplet radius will set in. By eliminating the need to extract the specimen from a mould, it prevents the interface from any mechanical stresses ensuring an undisturbed diffusion. The drops allow a radial symmetric look on the interface diffusion, allowing a one-dimensional modulation. Although the impact of volume diffusion has to be accounted for by adding a second dimension, it is mainly assumed to take place perpendicular to the interface. 

Silicone rubber was prepared at a 10:1 volume ratio of resin to curing agent, mixed thoroughly and degassed several times. Silicon wafers (3 inch) were used as substrates. They were cleaned via inverse sputter etching of about 4 nm (ISE-90, von Ardenne Anlagentechnik, Germany) at a power of 200 W and a working pressure of 8 µbar. Substrates were heated at 200 °C for 10 min. A Cr/Ni adhesion layer with a thickness of 4 nm was sputter deposited (PPS-90 UV, von Ardenne Anlagentechnik, Germany) at 400 W power, succeeded by 200 nm copper deposited at 500 W. Substrates were tempered at 200 °C for 10 min. Layer thickness was measured via white light interferometry and phase shift interferometry (PLU-neox, Sensofar, Terrassa, Spain).

In order to vary the surface energy leading to a different contact angle to investigate the impact of the edge geometry (starting point) on the interface diffusion process, some wafers were additionally vapour coated with silicon oil SF V50 at room temperature (“siliconization” in contrast to “silazanisation” with e.g., HDMS).

Directly after removal from the sputter system or after the siliconization process, respectively, several drops of 40 µL volume of the prepared PDMS were set with a cannula directly on the surface of the wafer. The PDMS was then polymerized at 40 °C for 48 h. The low temperature was chosen due to the different temperature coefficients of expansion of copper and the polymer, which would lead to thermal stresses in the interface. Moreover, the process at 40 °C lowered the viscosity of Sylgard-184 leading to a further expansion of the edge of the drop. This process regime resulted in drops with a diameter of 3 cm by using only volumes of 40 μL PDMS. The number of samples prepared for this study was low due to ongoing COVID working restrictions.

Before further diffusion experiments, the drop geometry was investigated, with particular attention paid to the edge geometry, since it is considered the starting point of the diffusion process. For the edge-characterization, a thin layer of platinum was applied via sputtering to provide a better image in the light interferometer.

Light microscopic images show the geometry of the edge of the droplets, indicating a homogeneous surface energy in every point. Therefore, a constant contact angle was assumed for the complete edge of the droplet. This assumption was also applied for the contact angle, which was measured for the siliconized specimen. For both surfaces, an average contact angle of 5° ± 1° was determined. Nevertheless, high resolution images of the edges of the droplets on different surfaces (Figure 4) revealed a deviation in slopes close to the edges. While the slope for the siliconized sample continuously decreases towards 0°, the slope of the bare Cu surface drops down with a steeper slope. This is due to the better compatibility and thus, wettability of the siliconized copper surface as the chemical properties of PDMS and silicon coating on copper are quite similar. 

The edges of the specimen appeared smooth, while at higher magnifications (Figure 5, right) a wavy structure in the outer edge was revealed. It can be assumed that this microscopic roughness does not affect the interfacial diffusion process in our macroscopic view. Furthermore, in this picture, the formation of a 5 µm wide, thin film emerging at the outermost edge of the droplet can be observed. The formation of this zone can be explained by a segregation process of PDMS. In order to obtain the specified viscosity, oligomers with different molecular lengths were mixed. The miscibility of polymers depends on their molecular length and decreases with increasing chain length. This effect can be enhanced by increasing the temperature. Thus, the shorter, more mobile chains migrate more to the surface or interface, resulting in the thin film seen in the image. For silicone coating, PDMS oligomers are present at the surface. These are more compatible with molecules leaving the PDMS droplet and can enhance the spreading of the low molecular weight PDMS molecules, resulting in the continuous decrease in slope. 

First corrosion experiments in selective points on the edge show how corrosion leads to delamination after minutes (Figure 6). We observe three areas: Area 1 is the reagent exhausted proportion where the corroding agent has lost its full chemical potential. Area 2 shows full corrosion. In Area 3 we are already able to observe the incipient delamination. 

### 2.6. Specimen for Volume Diffusion

The volume diffusion through PDMS towards inner metal parts was studied with cuvettes imaged in a time lapse series (Figure 7). Small square platelets cut out of a copper-coated circuit board were placed in a polystyrene cuvette and doused with Sylgard-184. The platelet dimension was chosen to provide a natural slant within the cuvette. In this way, the material thickness of the PDMS layer above the platelet was varied. In addition, this slant allowed a lateral top view of the copper plate without having to look through the potassium-polysulphide solution, which is advantageous for a long-term experiment because the corrosion solution forms an opaque wax-like layer on the surface after a certain standing time. The cuvettes were then filled with the potassium polysulfide solution and imaged with the time-lapse camera (PIXEL LINK). The time-lapse images showed an onset of corrosion at the upper edge, after 1500 min. Within the next 1500 min, the entire copper plate blackened. With this series of experiments, the time of onset of volume diffusion, as well as the relationship between coating thickness and diffusion time, were determined. From the geometric position of the copper platelet within the cuvette, the thickness of the material above was determined. From the angular position of the copper platelet, the trigonometric ratio was used to determine the opposite cathetus and, subsequently, the height of the PDMS layer at the respective location. The layer thickness resulted from the sum of the opposite cathetus and an offset of the layer height lying above the edge. This layer thickness was viewed as a membrane through which potassium polysulfide ions permeate from the solution to the copper. The cuvette examined had an offset of 1 mm, while the copper plate sat at an angle of 30°.

## 3. Results

Interface diffusion in the metal-PDMS junction was observed within 120 min after the start of the experiment. The diffusion front parallel to the surface advances along the gradient of concentration much faster than in bulk. Because of the delayed start of bulk diffusion, we observe an isolated view on interface diffusion. With the chosen set up, bulk diffusion could be neglected. The diffusion front progresses over time for the case of undisturbed interface diffusion as shown on Figure 8. In a timeframe of minutes, the covered distance of the diffusion front can be measured to several tenths of a millimetre, indicating that the diffusion at the interface is up to 10 times faster than in the bulk of the material. When comparing different surface modifications, the specimen without a Cr adhesion layer (Figure 7, green curve) shows a linear increase in the curve, indicating an immediate onset of corrosion-induced delamination. In contrast, for the specimens with a Cr adhesion layer, the slope of the curves decreases. The velocity of diffusion correlates with the adhesive bond between the two components of the interface. A weaker bond at the interface (siliconized surface) leads to advanced diffusion velocities. 

Figure 9 shows the progression of the diffusion front for three samples. For three specimens, a similar type of progression of the curves can be found. All curves show a jump after about 420 min. Furthermore, curves can be divided into three characteristic segments. In phase I, the curve exhibits a saturation-like behaviour as the process of interface diffusion is mostly undisturbed and not troubled by other overlapping processes. In phase II, the adhesion of the PDMS to the copper has been compromised leading to separation of the two junctions. This marks the beginning of delamination which forms new cavities, allowing the corroding agent to flow under the PDMS leading to a jump in concentration. The saturation-like behaviour of the curve converts into an increasing slope, showing a faster progression of the corrosion after the beginning of delamination. Our investigation method reaches its limit in phase III, as the bulk diffusion becomes a factor. At this point, the remaining not yet corroded copper surface begins to corrode uniformly due to the bulk diffusion perpendicular to the direction of interface diffusion. This leads to a colour change and the contrast between the two states (fully corroded and non-corroded copper) decreases. The MatLab script used to evaluate the data is no longer applicable resulting in the sharp jump we can observe in phase III. 

### 3.1. Phase I

In the first segment, we observe a saturation curve-like progression. The front of diffusion penetrates the interface in a radial direction and corrodes the copper. This way, with progressing time, the diffusion flattens the concentration gradient along its way as the bonded corroding ions are immobilized on the copper surface and do not participate in a further diffusion process but inhibit it. With a greater travelled distance by the front of diffusion, the gradient flattens leading to saturation-like behaviour. This observation was transferred into discontinuous diffusion coefficients in our mathematical model as they change discontinuously from a constant value to another at certain concentrations. So, we modelled a moving boundary between the two phases of corroded and not-yet corroded copper, separating the region where all diffusing molecules are immobilized from the area in which they are allowed to diffuse freely as the concentration is still zero. With further progression of the diffusion, the boundary moves alongside the diffusion front. 

Mathematically, these kinds of boundary value problems are often referred to as Stefan-like. Stefan [25] used these special boundary conditions to model the melting process of an ice cube. During the melting process two phases can be observed, the liquid (melted ice) and solid phase. As time progresses, so the boundary between these phases moves. The growing liquid phase acts as a heat shield for the yet non-melted solid phase, slowing down the melting process. As the laws of Fick were derived from the heat equations, the Stefan problem is easily transferable to our diffusion problem [26]. 

In his work, “the mathematics of diffusion” Crank [26] transfers the Stefan-like behaviour to the problems of diffusion. According to Crank “formal mathematical solutions have been only obtained for a limited number of problems, mostly infinite or semi-infinite media” [26] whereas other problems, such as concentration dependent coefficients, need some approximation in a numerical approach. 

The experimental setup is able to visualize the position of the moving boundary at any time. This experimental result has been used later to validate the outcome of our theoretical solution. According to Crank and Hill, the boundary conditions suitable for our problem are: (1)$$\begin{array}{c} C=0,\;t=0,\;x>0\\ l\frac{\partial C}{\partial t}=D\frac{\partial C}{dx},\;x=0,\;t\geq 0\\ S\frac{dX}{dt}=-D\frac{\partial C}{\partial x},\;x=X,\;t>0 \end{array}$$

These conditions have to satisfy Fick’s Second Law of diffusion:(2)$$\frac{dC}{dt}=D\frac{d^{2}C}{dx^{2}}$$

In order to solve these problems, mostly approximated analytical approaches are made. Crank suggests a polynomial approach for the unknown concentration profile to satisfy all boundary conditions. 

With Goodman’s integral method [27] a polynomial approach for the concentration profile C is made to satisfy the conditions of the moving boundary with the constants a and b determined from the boundary conditions, x, any point in the material and the moving boundary at time point t is X (see equation set 1):(3)C=a(x−X)+b(x−X),

This leads to the solution of the position of the boundary according to time with X again the moving boundary, *t*: time and α a factor including diffusion coefficient, consumption rate and boundary concentration of Potassium polysulphide.
(4)$$X=\alpha \sqrt{t},$$

The position *X* describes the boundary in the photographs in which copper corrodes and turns from reddish into black. The square root dependency is consistent with our experimental data.

### 3.2. Phase II

The experimental set up allows two values of concentrations to be observed. One actively (Figure 10), one passively (Figure 11). The concentration observed actively—stage one—is the indicator concentration $C_{i}$ at which the copper surface visually begins to corrode and its colour changes to black. It indicates the position of the diffusion front (the moving boundary) and is mathematically described by Equation (4).

After a certain time, the second value of concentration gains significance. When the copper surface is corroded to a certain amount, the metal-polymer bond at the interface is compromised and the PDMS begins to delaminate. Because of the already blackened copper it is not possible to actively observe this process. 

Due to capillary forces, the corrosion-triggered delamination allows the corroding agent to migrate under the delaminated edge of the droplet leading to an increased jump in concentration. In this phase, a steady-state-like diffusion process can be assumed because the not-yet delaminated PDMS between the edge of delamination and diffusion front acts like a membrane. This steady-state behaviour is reflected in the almost linear part of the curve, which the diagram begins to evolve into in phase two (Figure 9). We hypothesize from this observation that the point of delamination correlates with the point of the diffusion front as it follows it in a constant distance. This allows for the assessment of the state of delamination and therefore the degradation in medical implants consisting of a polymer-metal composite.

### 3.3. Phase III

The evaluation of the time lapse series of the bulk diffusion experiment (Figure 7) led to the graph depicted in Figure 12, which, for the time interval 1500–2100 min, suggests a linear behaviour. After a time lag, the indicator concentration reached the upper edge of the copper plate at 1500 min. Measured data allowed for the approximation of the diffusion coefficient for the $K_{2}S_{x}$ ions in Sylgard-184 using the Einstein–Smoluchowski Equation (5)
(5)$$D=\frac{\Delta x^{2}}{2t_{\Delta x}}$$
and calculating the average diffusion coefficient to
$$\bar{D}\approx 1.67609\cdot 10^{-6}\;\frac{{\mathrm{cm}}^{2}}{s}$$

This result for the diffusivity of $K_{2}S_{x}$ ions in Slygard-184 lines up with other diffusivities of other molecules in Sylgard-184 [9]. Although that study investigated diffusion of gaseous molecules, while ours dealt with diffusion out of a liquid phase, the diffusivities are in the same order of magnitude [9] (Table 1).

As a final effect in phase III, the volume diffusion will meet the metal interface and initiate a fast and global weakening of the polymer-metal bond. This speeds up the interface diffusion and delamination, which then seems to be almost simultaneous over the entire remaining area when compared to the time scale of the preceding processes (Figure 13).

### 3.4. Influence of Surface Modification

The influence of surfaces modification in delamination due to interface diffusion was revealed (Figure 13). For the specimen without siliconization, a lack of adhesion due to higher surface corrosion and a faster delamination onset was found. It is visible as a jump in the plot at about 425 s which leads to a more rapid transition in the linear-like behaviour in Figure 13. In contrast to this, the siliconized sample reveals a nearly linear increase in slope with a small jump at the delamination onset, indicating higher adhesive strength of the polymer-metal bond. This is a strong indication that the diffusion speed directly correlates with the strength of the adhesive bonds of the interface. Meaning weaker adhesion leads to faster delamination resulting in an accelerated diffusion rate due to the jump in concentration on the new formed edge of the droplet. 

## 4. Discussion

With advancing miniaturization in AIMDs, the relation between bulk and interface has changed and the long-time negligible interface gains more and more importance. To predict the failure mechanisms in the polymer-metal junction due to corrosion-triggered delamination, a better understanding of diffusion processes in the interface is needed. In Figure 14 it is shown, that the delamination precedent is a diffusion of molecules in the interface, leading to the corrosion of the metal, thus weakening the metal-polymer interface adhesion. The proposed method enables an easy optical investigation of the diffusion process by changing the electrode material from platinum to copper, which is easy to process and can be corroded in a small amount of time allowing an isolated view on the diffusion processes. In this way, it is possible to obtain results on different degradation states in considerably shortened periods of time and to accelerate lifetime testing. Since only the speed of corrosion is changed the results for diffusion of corroding ions are still transferable to regular AIMDs. 

Cochlear implants always show internal or external stresses depending on the fabrication method but do not experience permanent flow of blood and large continuous rhythmic movements as cardiac pacemaker electrodes do. Fabrication via moulding always exerts forces on the interface when removed from the mould influencing the adhesive bond. Laser fabrication yields thermal stresses in the crucial area directly in the three-phase boundary. The simple specimen layout and short-time method allows high reproducibility and number of test series. The fabrication method reduced the influence of external stresses on the specimen. The samples largely cancelled out these stresses, allowing the study of diffusion and delamination processes at an undisturbed interface. Overall, specimen design and fabrication method are in close correspondence to regular neural implants but also facilitate an unimpaired investigation of unimpaired diffusion and delamination in contrast to industrial production methods. 

The investigation methodology developed in this study allows life-time testing from start to total failure in only a few days. Compared to usual life-time testing methods this means a large acceleration. In this way, diffusion and delamination phenomena can be observed and modelled for different specimen layouts and surface pre-treatments in short periods of time. Whilst the chemical reactions on AIMD electrode metals strongly differ from the fast copper reactions with the reactive sulphur compounds of sulphur liver, the general nature of the chemical interface reaction diffusion, competing with body fluid ingredient species volume diffusion remains the same. The discovered temporal correspondence of both processes on the copper/PDMS/ sulfur liver gave us the opportunity to develop a mathematical model of the inherent interface degradation processes. Over the whole test procedure, we were able to identify three phases (see video in suppl. Mat.).

### 4.1. Phase I

The developed model for the first phase shows good compliance with the isolated process of interface diffusion. This model, developed as a Stefan-like moving boundary problem, is able to theoretically describe the interface diffusion process. The model agrees with the experimental values for phase one gained in our experiments, making it the first model to predict the movement of the diffusion front in the metal-polymer-interface. There are quite a few studies for interface diffusion lacking a mathematical point of view. The model developed in this study is based on the already existing diffusion mathematics formulated by Crank, which has already been validated and proven over decades.

The model in this study uses Fourier’s laws of heat conduction from which Fick’s laws of diffusion were derived. Boundary conditions were inspired by the Stefan-problem, which characterize the melting or solidification of an ice cube. The one-dimensional model by Theodore R. Goodman describes a distinct melting line between the solid and liquid phase. This heat balance integral method, published in 1957, was subsequently used to describe several technical systems. For several cases, this method delivers viable solutions with minor numerical effort. Examples for applications are: heating by forced convection, heat conduction by transpiration, security analysis for nuclear reactors and also concentration-based diffusion processes [26,27,28,29]. 

For polymers, the Stefan-problem is used to describe pyrolysis. The polymer is exposed to thermal radiation, increasing the temperature up to a critical temperature leading to pyrolysis. A pyrolysis front (moving boundary) is formed, which can be characterized by the Stefan problem [30]. 

The equation for the position of the front of diffusion is quite simple showing a square-root-character (Figure 15). We also found a direct correlation between the position of the diffusion front and the state of delamination leading to phase two.

### 4.2. Phase II

With the knowledge of the correlation (Figure 16) between diffusion and corrosion-triggered delamination, it is possible to predict the position and state of delamination in the polymer-metal-junction. This allows estimating important causalities in degradation leading to material failures. With these understandings, new ways of testing and highly accelerated lifetime testing (HALT) are possible. Before this, life testing methods were heavily based on increasing factors of temperature and electric current in amperage and frequency. With better knowledge of the processes of interface diffusion and delamination, new factors of HALT open up [23,31,32]. Furthermore, a first estimation in ratio of diffusion rate between bulk and interface could be made marking interface diffusion up to 5 times faster than in bulk. In general, the diffusion speed of the used ion compounds can be identified as quite high. 

### 4.3. Phase III

Slower diffusion in bulk allows an isolated view on the interface diffusion/delamination process in the starting phase. The results of the bulk diffusion experiments show that for further modelling, a more accurate study for the bulk diffusion is needed to take its impact in phase III into account. At this time, this represents the limit of our current model and further investigation is required.

The developed model is still based on and therefore still attributable to the preceded and validated maths of Crank [26], Goodman [27], Gupta [33] and Stefan [25] lining up the interface diffusion process with other standard moving boundary condition problems. The model was validated for our experimental data. Two of the three phases could be modelled and mathematically described. Phase one represents the undisturbed interface diffusion, whereas phase two represents the correlating superimposed processes of interface diffusion and delamination. For this second phase we postulated an approach of explanation with the possibility of further mathematical modelling. For the third phase, with bulk diffusion as the dominating process, additional investigations are necessary in additional studies.

## 5. Conclusions

Although copper is not a standard material in medical implant technology, the revealed separate interface and volume mechanisms found can be further generalised to understand delamination failure between polymers and AIMD electrode metals such as platinum and platinum-iridium alloys. The interfacial diffusion with mobile phase boundaries according to the Stefan model and the diffusion of constituents of the body fluids, which leave the aqueous phase and cause damage over large areas, applies to all PDMS-metal interfaces. Although the chemical reactions of platinum metals occur at different potentials, damage is certainly observed under the usual medical stimulation conditions, e.g., in CIs. For their analysis, however, waiting for visible damage could be replaced by faster and more informative methods. Such as confocal RAMAN microscopy, for example, which works in the transmissive window of PDMS, would be suitable for this. It allows the identification of reactants, opening the possibility of detailed material parameterization for a delamination model that could support the development of digital twins of failure modes and thus improve AIMDs in the future.

## Figures and Tables

**Figure 1 bioengineering-09-00010-f001:**
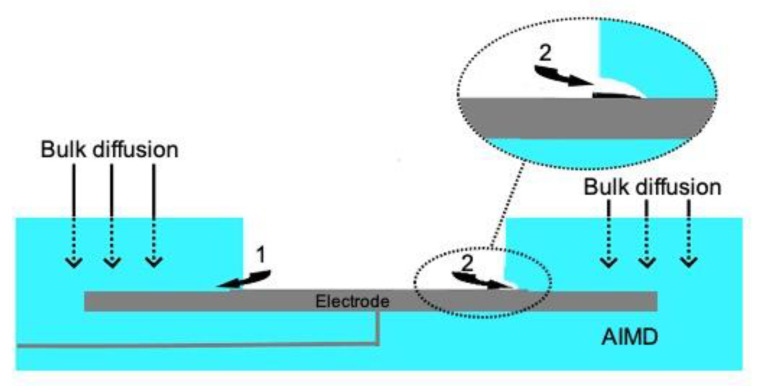
Two processes accelerate delamination in an AIMD: Initially, interfacial diffusion occurs along the metal-polymer interface, which initiates corrosion of the metal surface (**1**) Corrosion leads to delamination between the metal surface and the PDMS (**2**), creating a cavity shown in detail in the enlargement of (**2**). Bulk diffusion additionally intensifies the corrosion process.

**Figure 2 bioengineering-09-00010-f002:**
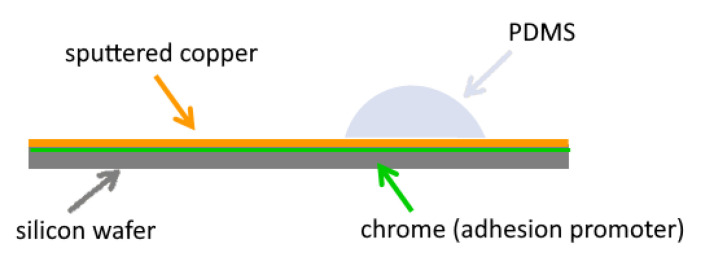
Specimen layout. A silicon wafer (with chrome as optional adhesion promoter) is coated with copper via PVD. On this copper surface (further pre-treatment possible), drops of Sylgard-184 are placed and polymerized.

**Figure 3 bioengineering-09-00010-f003:**
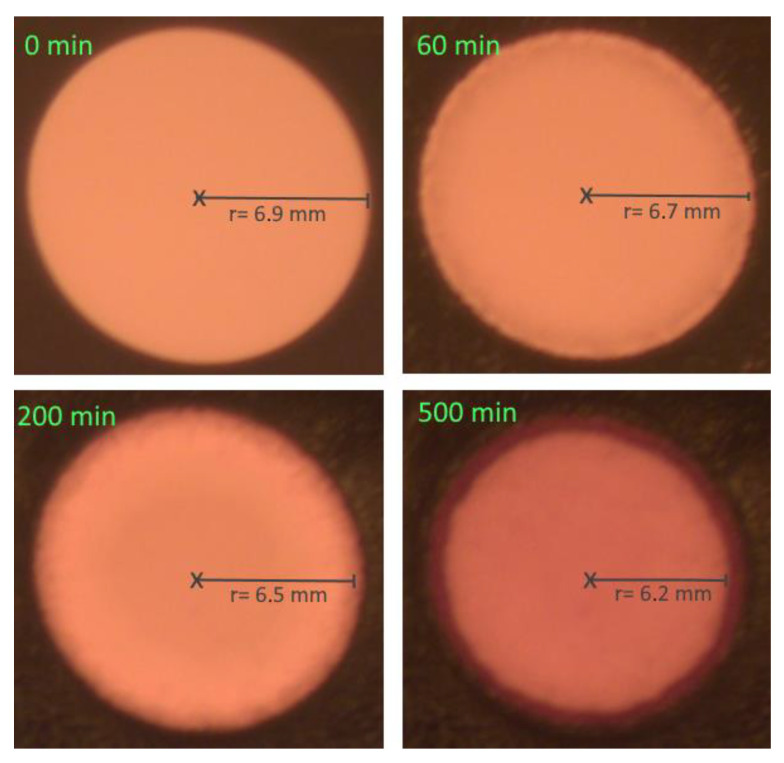
The specimen immersed in the corroding agent. The exposed copper surface immediately begins to corrode (brown/black), while the PDMS drop protects the underlying copper (red/orange). Pictures show the advancing corrosion at different time stages. With full corrosion of the copper the radius r of the drop begins to decline due to the corroding agent penetrating the interface and corroding the underlying copper.

**Figure 4 bioengineering-09-00010-f004:**
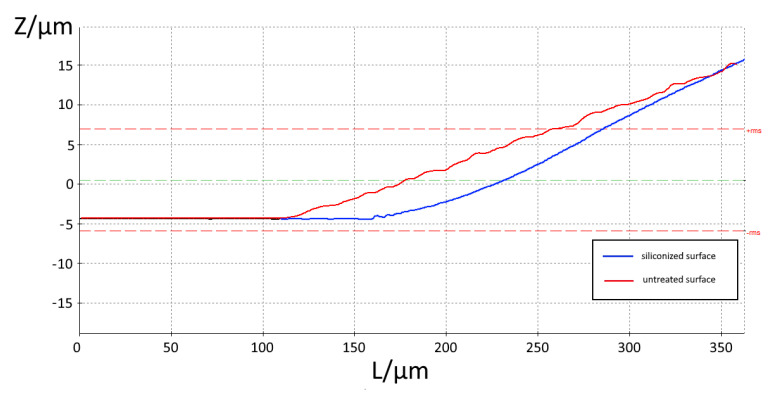
The profile of the droplet edge (L: length; Z: height of the droplet) shows a correlation between the surface composition and the profile of the droplet in the edge region. Due to the better wettability of a siliconized copper surface, the contact angle converges at the outer edge and forms a thin film already visible in the microscopic image (cf. Figure 5).

**Figure 5 bioengineering-09-00010-f005:**
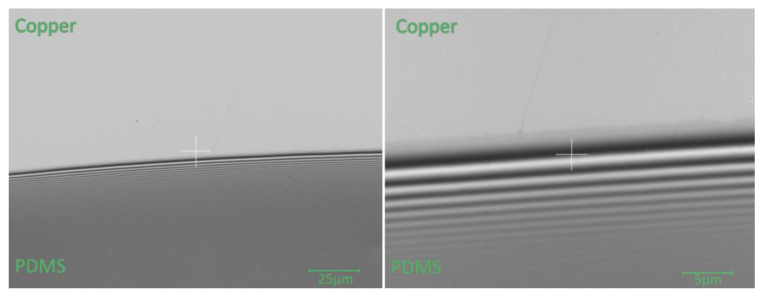
Edge of the PDMS droplet on siliconized Cu exhibits a homogeneous and sharp edge (**left**). At higher magnification (**right**), a thin film can be seen emerging from the edge. This film is assumed to have negligible influence on the macroscopic view of interface diffusion.

**Figure 6 bioengineering-09-00010-f006:**
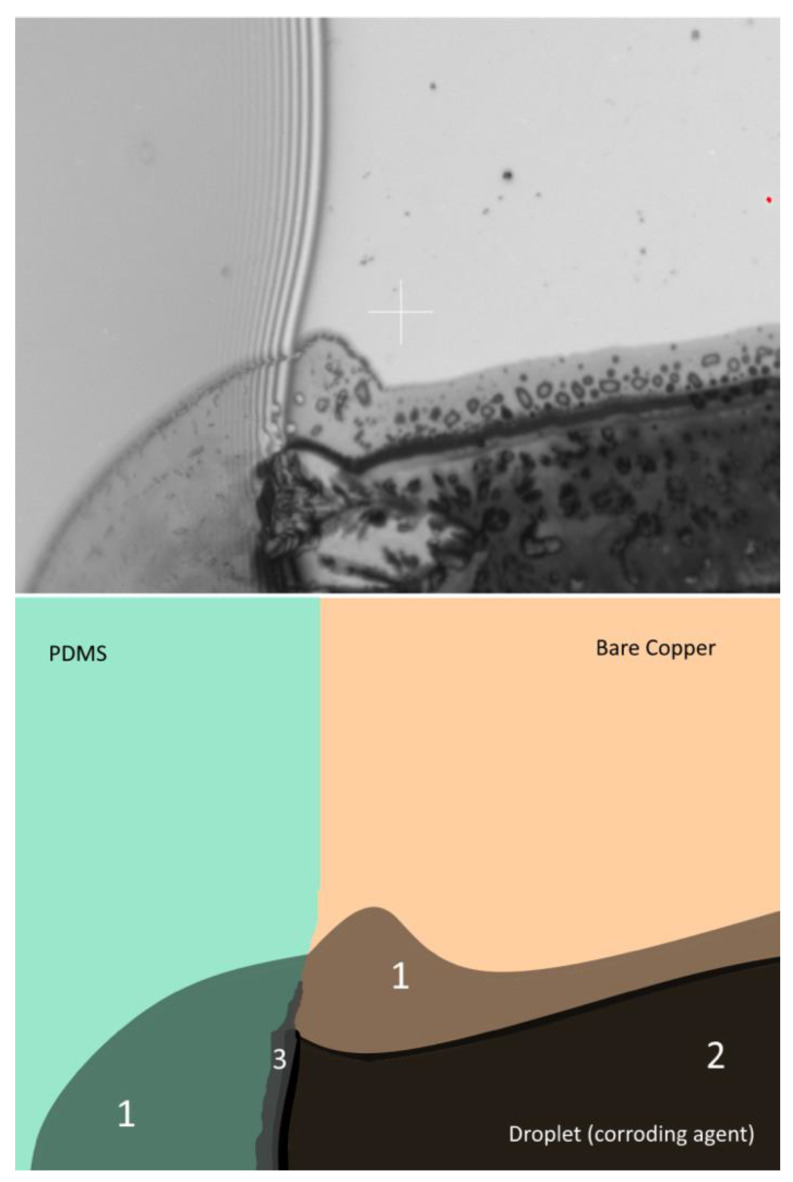
Forming of a multi-phase at selectively induced corrosion. The edge of the PDMS drop was partially exposed to a small amount of the corroding agent. The picture was taken 10 min after exposure to the corroding agent. Area 1 shows the area of depletion of the corroding agent. The concentration is not sufficient to corrode the copper fully. Area 2 displays the area of full corrosion. Area 3 shows the beginning of corrosion-triggered delamination, which becomes visible as a darker area as the underlying copper corroded to a certain degree where the adhesion in the metal-PDMS-joint is compromised.

**Figure 7 bioengineering-09-00010-f007:**
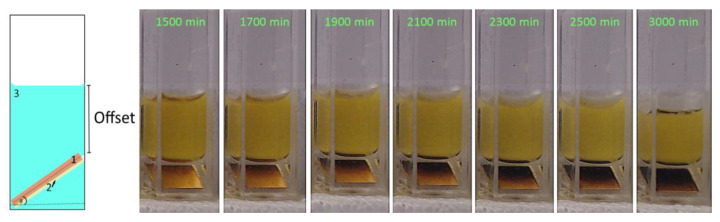
Course of the bulk diffusion experiment. After 1500 min, the indicator-plate begins to blacken, confirming a certain concentration has reached the upper edge of the plate. After 3000 min the critical concentration reaches the bottom of the cuvette.

**Figure 8 bioengineering-09-00010-f008:**
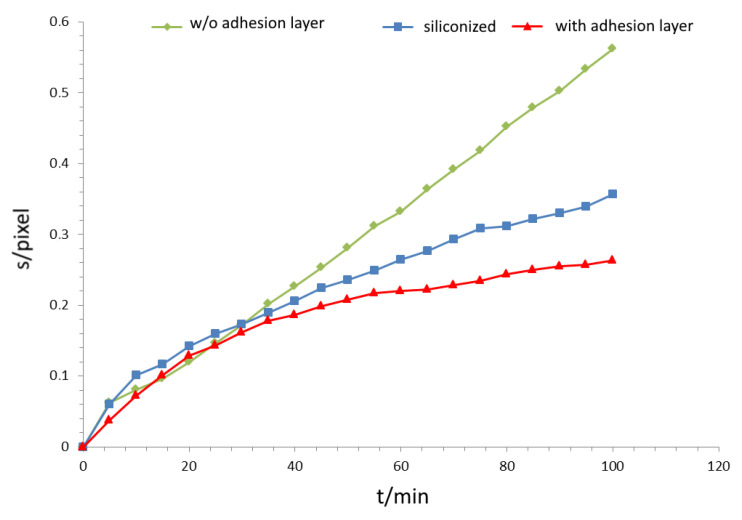
Diagram of interface diffusion (Phase I) for three samples with different surface modifications. The green curve displays the behaviour of a silicone/Cu sample without an adhesive layer. Here, the corrosion-triggered delamination sets in almost immediately converting in a linear course. The other curves represent a substrate with adhesion layer. Both curves show a similar behaviour with a slightly faster diffusion in the siliconized sample due to the flatter edge of the drop but slower than the blue one. The unit “pixel” for the length of s is the output of the MatLab algorithm.

**Figure 9 bioengineering-09-00010-f009:**
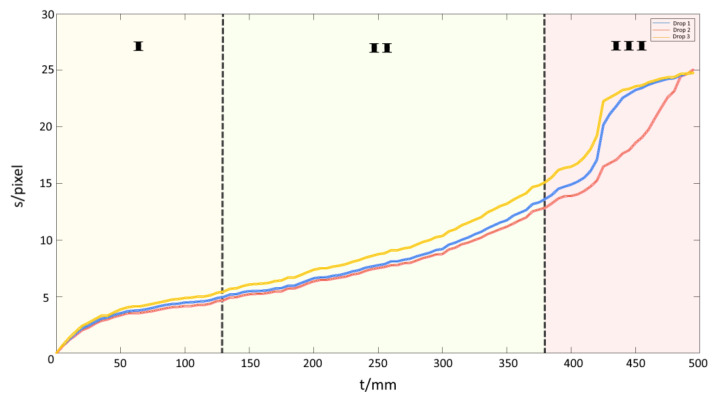
The three phases of the corrosion process (three samples given in red, blue and yellow curves). Phase I: Shows the saturation-like behaviour of interface diffusion. In this stage, interface diffusion can be investigated as an isolated process. Phase II: Marks the beginning of delamination of the droplet’s edge. The corrosion has progressed so far that the adhesive bond is compromised to a point that the PDMS begins to separate from the surface. The corroding agent flows into the newly formed cavity leading to a jump in concentration. This effect leads to a conversion into a linear behaviour. Phase III: Bulk diffusion sets in and corrodes the copper uniformly inhibiting any further interface diffusion. Furthermore, it marks the limit of our investigation method as MatLab is not able to differentiate between corroded and non-corroded copper by colour. The visible jump in state III is due to the threshold, which is not applicable anymore. The unit “pixel” for the length of s is the output of the MatLab algorithm.

**Figure 10 bioengineering-09-00010-f010:**
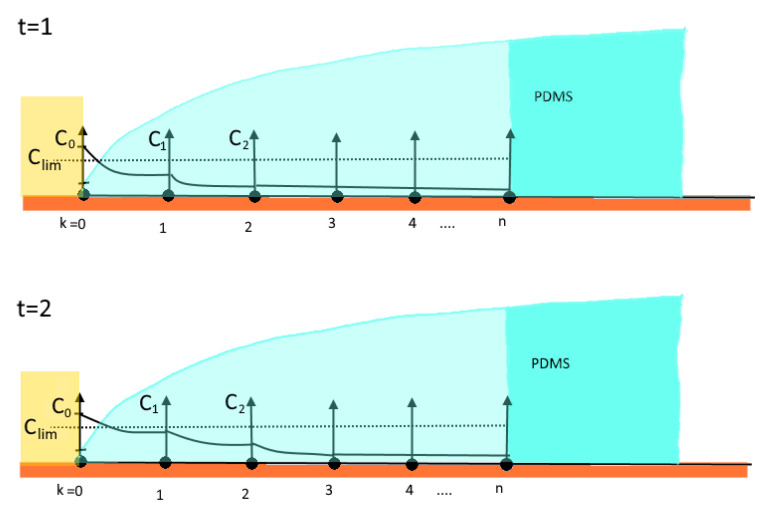
The beginning of the interface diffusion. The diffusion range is discretized in finite elements. At *t* = 1 the interface diffusion is at the starting state, only affecting the first element. In the second stage, due to the coupling of the nodes, the concentration begins to spread into the next element. Corrosion in the first element has progressed so far that in *t* = 3 the PDMS begins to delaminate from the copper substrate. In this way, a cavity is formed allowing the corroding agent to flow under the separated PDMS. Furthermore, the corroding agent directly affects the next element leading to a jump in concentration. This marks the beginning of the conversion of the curve from a saturation-like shape into a linear-like behaviour.

**Figure 11 bioengineering-09-00010-f011:**
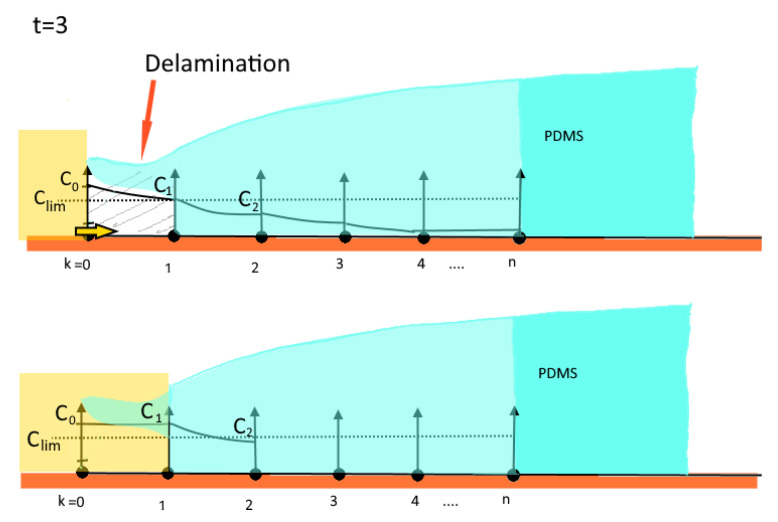
Corrosion in the first element has progressed so far that at *t* = 3 the PDMS begins to delaminate from the copper substrate. In this way a cavity is formed allowing the corroding agent to flow under the separated PDMS. Subsequently, the corroding agent directly affects the next element leading to a jump in concentration. This marks the beginning of the conversion of the curve from a saturation-like shape into a linear-like behaviour.

**Figure 12 bioengineering-09-00010-f012:**
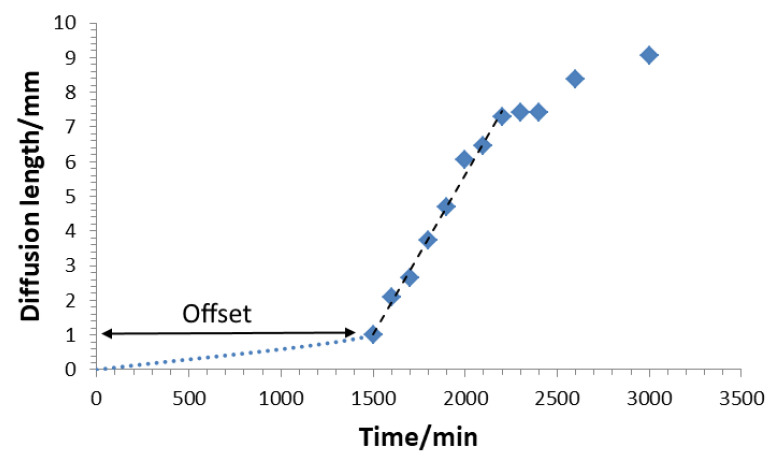
Results of the diffusion length of bulk diffusion. After a plateau phase where no colour change is visible on the indicator plate, the effect of bulk diffusion becomes visible. The offset is the time the corroding solution needs to diffuse through the bulk of the PDMS to reach the copper plate. During the time interval from 1500 min up to 2300 min, the corrosion of the copper migrates linearly along the tilted plate. Towards the end a saturation phase was observed as the gradient of concentration decreases.

**Figure 13 bioengineering-09-00010-f013:**
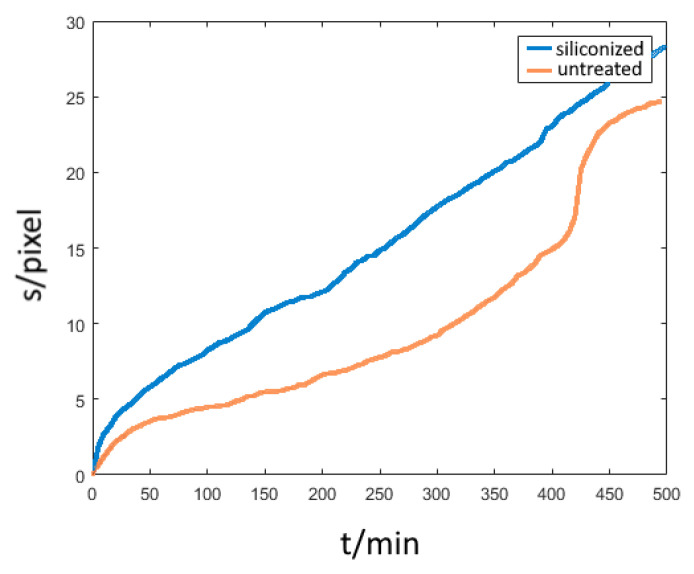
Influence of surface treatment on interface diffusion using a thin PDMS film. The adhesive bond between Sylgard-184 and the siliconized copper is weaker, thus leading to faster corrosion-triggered delamination reflected in the more pronounced linear behaviour. Furthermore, the shallower and thinner edge accelerates the diffusion in the starting phase.

**Figure 14 bioengineering-09-00010-f014:**
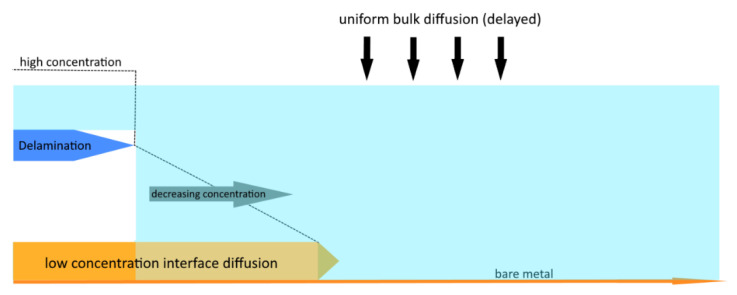
Overview of different processes of implant degradation from the outer rim (left) and via the volume (droplet top). At one point interface diffusion and bulk diffusion will merge and lead to an almost simultaneous, global delamination.

**Figure 15 bioengineering-09-00010-f015:**
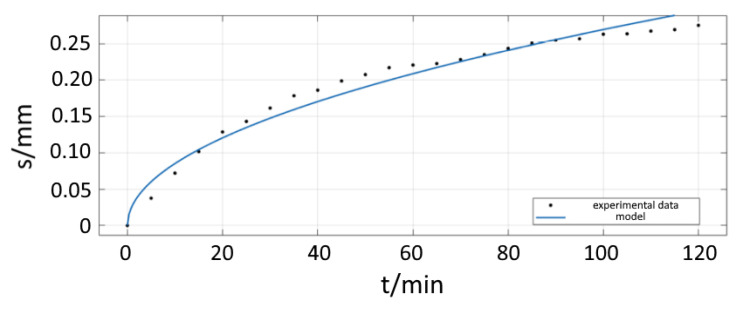
Model for phase I of interface diffusion.

**Figure 16 bioengineering-09-00010-f016:**
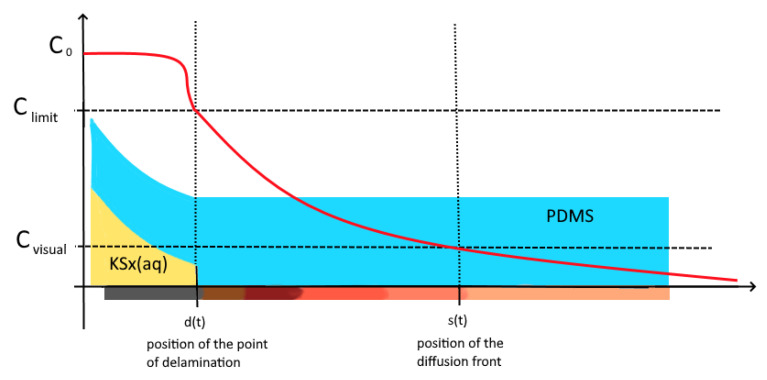
Correlation between the point of delamination and the position of the diffusion front. The delamination is assumed to lag behind the diffusion front at a constant distance. Therefore, a prediction of the state of delamination is indirectly possible by the knowledge of behaviour of the diffusion front. The boundary condition for the concentration of $K_{2}S_{x}$-Ions in the corroding agent is described by $C_{0}$. This boundary condition moves with the position of the delamination front d(t) over time. The concentration at which the PDMS begins to delaminate due to the advanced state of corrosion is described by $C_{limit}$. $C_{0}$ marks the concentration at which the corrosion is visually observable by colour change.

**Table 1 bioengineering-09-00010-t001:** Molecule diffusivities in Sylgard-184 [27].

Gas	Sylgard-184
Water vapor	$5\times 10^{-6}$ cm^2^/s
$H_{2}$	$1.4\times 10^{-4}$ cm^2^/s
$N_{2}$	$3.4\times 10^{-5}$ cm^2^/s

## Data Availability

We will prepare the original diffusion data in repository on our website.

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
