# Peer review of "Predicting Corrosion Delamination Failure in Active Implantable Medical Devices: Analytical Model and Validation Strategy"

_bioengineering, 2021, doi:10.3390/bioengineering9010010_

Round 1

Reviewer 1 Report

  • There are several minor spelling errors/typos throughout the manuscript. Please proof-read again to fix.
  • Furthermore, it is customary for academic papers to exclude the use of first-person point-of-view (e.g., ‘I’, ‘We’) in the manuscript sentence structure. Please amend the text to indicate this requirement.
  • Section 2.0 does not clearly discuss either the materials or methods used for your study. The section text should be moved to the introduction as it is more closely discussing the rationale behind your study
  • For section 2, please provide all tool manufacturer and model/part #’s for all metrology and processing steps used. Also please provide any important parameters used for the tooling/deposition steps (e.g., sputter temperature/pressure) and what metrology was used to determine film thickness (e.g., 200 nm Cu).
  • Please define chrome and platinum deposition thickness. A ‘thin’ layer is not that descriptive and not quantitative at all.
  • Silanization method pg 5, line 203 needs to be described. Details to include: silanization type (e.g., aminopropyltriethoxy silane, etc.), deposition method (e.g., vapor deposition, etc.), and post-processing (e.g., 105 ËšC thermal cure, etc.)
  • For figure 4, why does the baseline of the Z-height (at the left side of the plot) not start at Z-height=0? Is there a reason that you chose for it to start at -4.5 um (estimated)
  • For pg 6, line 225, how was the 5Ëš contact angle determined and how are you defining this angle (e.g., angle formed between the Cu-PDMSe-air interface?). Was contact angle estimated from figure 4 data using a top-down view of the PDMSe droplet? Or was it estimated using a side-angle view of the droplet on the Cu surface. Furthermore, please provide additional details for how the contact-angle of the silanized and non-silanized surfaces are the same. If you chemically altered the surface of the Cu, the angle the PDMSe formed on the Cu surface should be measurably different.
  • Pg 7, line 246-249, please provide additional details for how the corrosion experiments were performed and what observations/metrology tools were used during the experiments. For example, how much of the corrosion solution was applied? How was it applied (e.g., syringe)? Was the corrosion front over the Cu surface visualized at all time-points starting at t=0, were time-lapse snapshots taken, or was video recorded? What microscope/imaging system was used? What magnification? Etc.
  • Figure 6, please provide the timepoint that this observational image was taken
  • Figure 7, there are several issues that need to be resolved. 1) for the three curves, there are no error bars/standard deviation information and no information on sample size/# of replicates so the conclusions and discussions based off this information are meaningless/cannot be trusted. Please include this information. 2) The curve color labels (blue/grey/orange) do not match the figure. I see blue/red/green curves. 3) The definition of an ‘adhesion promotor’ is vague and misleading and needs to be clarified. Are you referring to the chromium layer used at the Si wafer-Cr-Cu interface? If so, how does that layer impact the Cu-PDMSe interface? Or are you referring to a previously undisclosed Cu-Cr-PDMSe interface? If so, please include the information in materials/methods on the application of this additional Cr adhesion layer on the Cu surface before PDMSe curing.
  • Pg 8, line 288-289, no details of this Matlab script were included in the materials and methods section of the manuscript. Please provide details on the development/functionality/use of this script for the data collection and analyses used.
  • Similarly, the mathematical methods and development of the mathematical model were not included in the methods section. Please include these details
  • Figure 8, please define what the three curves indicate. The legend ‘drop 1’, ‘drop 2’, etc. is unclear. Are these simply replicates? Or different surface treatments? If replicates, why not just show average behavior +/- error envelopes with proper discussion of statistical/analytical methods used?
  • Pg 9, line 314, is there a seminal paper or book that can be referenced from ‘Stefan’ to support the ‘stefan-like’ assertion that is made in the text.
  • Pg 10, please define the variables used for equations 1-2. It is unclear what each of the variables physically relate to and how this is correlated to figure 8 boundary line between stage I-II.
  • Figure 11, no methods were used to describe any ‘bulk diffusion experiment’ using cuvettes and is unclear what exactly the figure is trying to indicate. The sample architecture/setup is completely unknown, so the data discussion and conclusions drawn are confusing. Are the samples still Si wafers with Cu and PDMSe droplets? Or is the PDMSe fully coating the exposed Cu surface? Furthermore, the graphic on the left side of the figure showing the offset and angular dependency is not explained. Please rectify these concerns.
  • Figure 12, no explanation of the offset was included. Is this indicating that the time from t=0-1500 is all interfacial based diffusion? Or is this 1500 min simply the time it takes for the corroding solution to diffuse through the bulk of the PDMSs? If so, what is the thickness of PDMSe that was diffused through (e.g., what is ‘x’ in eqn. 3)
  • Pg 12, line 392. Figure 14 was discussed in the text before any mention of figure 13. Please re-arrange the figures/text as necessary to fix this discrepancy

Reviewer 2 Report

In this article, the authors used a model experimental system to visualize and study the delamination between noble metal and polymer. Using interface diffusion, the authors developed mathematical models to quantify the propagation of diffusion front and correlate the corrosion with delamination. The results are thorough and interesting. However, there are some problems in the general description of the experimental method and interpretation of the results observed. The paper also needs careful proofreading before it can be accepted.

Major comments:

  1. The delamination in AIMDs is a complicated result from both overstress and wear-out mechanisms and corrosion is only one chemical reason that belongs to the chemical wear-out mechanism (refer to Figure 7.1 in the book “Adhesion in Microelectronics”, DOI: 10.1002/9781118831373.ch7). The author seems to use corrosion to indicate the delamination process, which is not correct. It is suggested for the authors to change “delamination” to “corrosion” in their manuscript.
  2. Copper is not safe for clinical devices. The authors should comment on whether the conclusion discussed could apply to gold or platinum electrodes, especially since these metals won’t be etched easily, which is fundamentally different from the copper experiment.
  3. The model developed here is in a static environment. While for AIMDs, they are usually embedded in an actively moving environment, due to muscle contraction or blood pumping in the vessels. Thus, the mechanical response mismatch between different materials might become more significant in causing delamination. In this sense, the authors are suggested to carefully comment on the difference between their model system and practical devices.

Minor comments:

  1. On line 75, “mayor” should be major.
  2. Missing references. On line 94, “works of Crank and Goodman” should be referenced.
  3. On line 159, “solved” should be dissolved.
  4. Label incorrect. On line 233, “Figure 14” should be Figure 5.

Reviewer 3 Report

There is a high number of ‘we’ sentences. Re-write is suggested. The passive structure could be better.

In Figure 1, it looks like 2 is the detailed view of 1. Clarification or redraw is suggested.

141 needs reference.

Top part of Figure 6 is a concern for this reviewer. Explain and label.

Figure 7 and 8 are confusing. s/pixel and s [pixel] are unclear.

307-320 has no depth.

Co, Climit and Cvisual are unclear in Figure 16.

Round 2

Reviewer 1 Report

 Thank you for your feedback and for the revisions made. There are several remaining points of confusion that need to be rectified.

1) Please define in clear terms in the text what the differences are for a 'siliconized' surface and an 'untreated' surface are. As far as I understand it, all corrosion tests were performed with a cured PDMSe droplet, and I therefore don't understand what a 'siliconized' surface is or how it different from an untreated Cu surface

2) Please include your rationale for the small sample size due to the COVID pandemic. That information will be useful for readers to understand the existing limitations and show value in replicating your work in their own labs

3)  The cuvette testing methods that you added to the results section should be moved to the materials/methods section of the manuscript. The results section should be limited to discussing results.

Author Response

Dear reviewer,

thank you so much!

1) Your first input of the second round helped us to clarify a possible misunderstanding: We used room temperature vapour coating of the entire copper wafers from a silicon oil source, which is a standard hydrophobization procedure. We called this "silliconization" to make clear, that we performed something different from classical "silanization" in semiconductor processing which includes siliazanes (e.g. HDMS). 

The new text reads now: In order to vary the surface energy leading to a different contact angle to investigate the impact of the edge geometry (starting point) on the interface diffusion process, some wafers were additionally vapour coated with silicon oil SF V50 at room temperature (“siliconization”in contrast to “silazanisation” with e.g. HDMS).

2) COVID lab closure: We added in line 244 " The number of samples prepared for this study was low due to ongoing COVID working restrictions."

3) We rearranged the volume diffusion experiment into the materials/experimental section by creating a subsection "2.5 Specimen for volume diffusion".

Reviewer 2 Report

Appreciate authors’ careful responses to my previous comments. The authors have addressed most of my concerns. Here are some additional comments for the authors to consider before the acceptance of the manuscript.

The author mentioned in their response to my previous comment 1, “In the case in which PDMS is placed on a metal –might it be an electrode site or a metal surface of a hermetic package to protect electronic circuitry, we see adhesion failure of PDMS on underlying metals, which is usually called “delamination” in the medical device field. Our aim of the study is to find an explanation of the root cause of delamination in this setting.” In this specific context, I agree that corrosion seems indeed the major reason for delamination. However, the author used “Comprehensive analytical model” and “Active implantable medical device” in their title, which is not accurate for the specific environment discussed in this model experiment. The problem is that by ignoring other critical contributing issues (e.g., mechanical, and electrical mechanisms), the model is no longer comprehensive for AIMDs. I would like to bring to the authors' attention that the whole field of flexible devices is largely motivated by the mechanical property mismatch between bulk electrodes and biological tissues. Thus, I would strongly suggest the authors modify their title to make sure that “corrosion-triggered” delamination is included.

Author Response

Dear reviewer,

We understand your objection and agree with it. The title was narrowed towards:

Predicting Corrosion Delamination Failure in Active Implantable Medical Device: Analytical Model and Validation Strategy

Thank you and best wishes,

Theodore Dpöö